# Stochastic Layer-Wise Shuffle: A Good Practice to Improve Vision Mamba Training

## Abstract

Recent Vision Mamba models not only have much lower complexity for processing higher resolution images and longer videos but also the competitive performance with Vision Transformers (ViTs). However, they tend to fall into overfitting and thus mainly reach up to a base size (about 80M). It is still unclear how vanilla Vision Mamba (Vim) can be efficiently scaled up to larger sizes, which is essentially for further exploitation. In this paper, we propose a stochastic layer-wise shuffle regularization, which empowers successfully scaling non-hierarchical Vision Mamba to a large size (about 300M) in a supervised setting. Specifically, our base and large-scale ShuffleMamba models can outperform the supervised ViTs of similar size by 0.8% and 1.0% classification accuracy on ImageNet1k, respectively, without auxiliary data. When evaluated on the ADE20K semantic segmentation and COCO detection tasks, our ShuffleMamba models also show significant improvements. Without bells and whistles, the stochastic layer-wise shuffle has the following highlights: (1) *Plug-and-play:* it does not alter model architectures and is omitted during inference. (2) *Simple but effective:* it can improve the overfitting in Vim training and only introduce random token permutation operations. (3) *Intuitive:* the feature token sequences in deeper layers are more likely to be shuffled as they are expected to be more semantic and less sensitive to patch positions.

## 1 Introduction

Vision Transformers (ViTs) have showcased powerful capabilities for sequentially modeling visual data (Dosovitskiy et al., 2021; Liu et al., 2021; Dong et al., 2022; He et al., 2022; Bao et al., 2022), but are plagued by quadratic complexity for sequence length (Katharopoulos et al., 2020). State Space Models (SSMs) (Kalman, 1960; Gu et al., 2021a;b; Smith et al., 2023) have recently gained traction as potentially efficient alternatives to traditional Convolutional Neural Networks (CNNs) and ViTs as sequence-based vision encoders (Zhu et al., 2024; Smith et al., 2023; Liang et al., 2024). Thanks to the hardware-aware property and flexible selective scan computation, Mamba (Gu & Dao, 2023) stands out in a group of SSMs. Compared to the quadratic computational complexity of Transformers, Mamba architecture can scale to longer sequences with only nearly linear complexity, thus has been adapted to the vision field as backbone models (Zhu et al., 2024; Liu et al., 2024b; Wang et al., 2024). The recent efforts have paid to exploring 2-D vision data scanning routes and incorporating visual priors into Mamba token mixers (Zhu et al., 2024; Li et al., 2024; Yang et al., 2024; Huang et al., 2024). These Mamba models are experimentally demonstrated to be competitive to the ViT family or their hierarchical counterparts while maintaining the sequential scalability advantage. Such models showcased superiority in both supervised pre-training and downstream tasks (Chen et al., 2024; Patro & Agneeswaran, 2024).

Nevertheless, issues still hinder the further application of Vision Mamba models. The overfitting and performance degradation plague the series of models to be scaled up further (Zhu et al., 2024; Yang et al., 2024; Li et al., 2024; Wang et al., 2024), which is essential for nowadays backbone networks. The successfully trained models are mainly at the base or even smaller size and thus are inferior to CNNs and ViTs in terms of model capacity (Liu et al., 2024b; Huang et al., 2024). On the other hand, various training techniques have been applied but still no satisfactory situation has arisen. A very recent Mamba-Reg (Wang et al., 2024) work successfully trained large-size Mamba models using registers to eliminate the impact of high-norm regions in features. Such a method needs to

introduce a group of extra tokens into the plain structure. It is still an emergency to explore how the vanilla Vision Mamba model can be scaled up.

In this paper, we argue that new training techniques should be proposed to mitigate the overfitting problem for scaling vanilla Vision Mamba (Zhu et al., 2024) up. Starting from the sequential computation of Mamba and positional transformation invariance, we present a *Stochastic Layer-Wise Shuffle training regularization algorithm* that successfully helps to improve the large-size vanilla Vision Mamba model training. Specifically, deeper layers are expected to be more semantically sophisticated and less sensitive to low-level positional information, while shallower units should be better at sensing initial input data. Consequently, our regularization includes a token shuffle procedure to enhance the positional transformation invariance, along with a layer-dependent probability assignment according to the layer perception assumption. As a plug-and-play algorithm, our method neither brings the heavy cost for training nor changes the Vision Mamba architecture. Ablation results demonstrate the effectiveness of our regularization for addressing overfitting and the computation efficiency. Additionally, the trained ShuffleMamba-L achieves up to 83.6% accuracy on ImageNet classification (Deng et al., 2009), 49.4 mIoU on ADE20K segmentation (Zhou et al., 2017), and even outperforms the ImageNet-21K pre-trained ViT on COCO detection task. These results reach the state-of-the-art place over the existing Vision Mamba models and outperform the similar-size ViTs.

## 2 RELATED WORK

**Vision Backbones**    In the field of computer vision, the exploration of efficient and scalable backbone architectures has led to significant advancements (He et al., 2016; Krizhevsky et al., 2017; Dosovitskiy et al., 2021; Zhu et al., 2024), primarily driven by CNNs (Simonyan & Zisserman, 2015; Li et al., 2019; Liu et al., 2022b) and ViTs (Dosovitskiy et al., 2021; Liu et al., 2021; Wang et al., 2021) recently. Initially, CNNs serve as the foundation and have evolved into deeper architectures, such as AlexNet (Krizhevsky et al., 2017), VGG (Simonyan & Zisserman, 2015), and ResNet (He et al., 2016). Various studies have introduced advanced operators, architectures, and attention mechanisms to improve the effectiveness of models such as SENet (Hu et al., 2018) and SKNet (Li et al., 2019). The continuous refinement of convolutional layers has resulted in architectures like RepLKNet (Ding et al., 2022) and ConvNeXt (Liu et al., 2022b), which offer improved scalability and accuracy. Despite significant advancements, CNNs primarily focus on exploiting spatial locality, making assumptions about feature locality, translation, and scale invariance.

The introduction of ViT (Dosovitskiy et al., 2021) marks a turning point. Adapted from the NLP community Vaswani et al. (2017), ViTs treat images as sequences of flattened 2D patches to capture global relationships (Liu et al., 2022a; Wang et al., 2021). As ViTs evolved, models like DeiT addressed optimization challenges (Touvron et al., 2021; He et al., 2022), while others introduced hierarchical structures and convolution operations to incorporate inductive biases of visual perception (Liu et al., 2021; Wang et al., 2021; 2022). These modifications allow for better performance across diverse visual tasks, although at the cost of added complexity in the models. Recently, there has been a trend of reverting to the original, plain ViT architecture due to its simplicity and flexibility in pre-training and fine-tuning across tasks (Bao et al., 2022; Xia et al., 2022; Carion et al., 2020; Cheng et al., 2022). However, one of the major challenges is the quadratic complexity of the self-attention mechanism (Katharopoulos et al., 2020; Zhu et al., 2023), which limits the number of visual tokens that can be processed, impacting scalability.

**State Space Vision Models**    Early state space transformations (Gu et al., 2021a;b; Smith et al., 2023; Gu et al., 2023), inspired by continuous state models and bolstered by HiPPO initialization (Gu et al., 2020), showcased the potential for handling extensive dependency problems (Nguyen et al., 2023; Tallec & Ollivier, 2018). To overcome computational and memory issues, S4 (Gu et al., 2021a) enforced diagonal structure on the state matrix, while S5 (Smith et al., 2023) introduced parallel scanning to enhance efficiency further. The Mamba model (Gu & Dao, 2023) stands out for its novel approach to SSMs. Parameterizing the state space matrices as projections of input data, Mamba proposed the more flexible selective scanning.

While ViTs and CNNs have laid a robust foundation for various visual tasks, Mamba offers a unique potential due to the ability to scale linearly with sequence length (Patro & Agneeswaran, 2024; Zhu et al., 2024; Nguyen et al., 2022; Lieber et al., 2024). S4ND (Nguyen et al., 2022) is the

pioneering effort to integrate SSM into visual applications. However, the straightforward expansion of the S4 model did not efficiently capture image information. This gap led to further innovations in hybrid CNN-SSM architecture, such as U-Mamba (Liu et al., 2024a). Recent efforts have sought to build generic vision backbones purely based on SSMs without relying on attention mechanisms (Zhu et al., 2024; Liu et al., 2024b; Li et al., 2024; Yang et al., 2024; Wang et al., 2024; Huang et al., 2024). Vision Mamba model, built by sequentially stacking Mamba blocks, has been shown to outperform ViT in both tiny and small model sizes. VMamba (Liu et al., 2024b) incorporated the hierarchical prior into Mamba to enhance adaptability for visual tasks. There are also some work exploring to refine the scanning method in Vim for visual data (Yang et al., 2024; Li et al., 2024; Huang et al., 2024; Chen et al., 2024). Nevertheless, Vims are stuck into issues like overfitting and only Mamba-Reg (Wang et al., 2024) successfully scale it up by introducing a group of registers in the supervised training.

**Training Regularizations** To improve the training and generalization of deep models, various regularization techniques have been developed over the past years. Normalizations (Ioffe & Szegedy, 2015; Ulyanov et al., 2016; Wu & He, 2018) are proven to be effective for speeding the convergence up, in which the Layer Normalization (Ba et al., 2016) and RMSNorm (Zhang & Sennrich, 2019) are popular in training of large models. The family of data augmentations (Cubuk et al., 2020; Hoffer et al., 2020; Yun et al., 2019; Zhang et al., 2018a) help to produce more robust representations and enhance performance. Stochastic depth and drop path (Huang et al., 2016; Larsson et al., 2016) drop the connection in the block level, which can not only overcome overfitting but also decrease the training cost. Weight decay (Krogh & Hertz, 1991; Loshchilov & Hutter, 2019) is commonly adopted for mitigating overfitting as well in a weight-penalizing manner. Besides, the earlier Dropout approach (Srivastava et al., 2014) introduces disturbance by dropping hidden units. They have played roles in various network training scenarios. Despite their benefits, these existing methods show limitations for Vim training and scalability. In this paper, we argue that new regularization should be considered to address the overfitting problem and scale Vim up.

**Shuffle Models** Random shuffling is not a common practice in the field of visual modeling as it can be seen as a disturbance for the original signal. In the existing related work, ShuffleNet (Zhang et al., 2018b) proposed to shuffle channels on group convolution to design lightweight CNN. Spatially Shuffled Convolution (Kishida & Nakayama, 2020) designs a permutation matrix for input spacial shuffling to enhance the receptive field perception of convolution. Besides, Shuffle Transformer (Huang et al., 2021) introduces the shuffle operation across different windows for hierarchical Transformer models with the motivation of improving the long-range vision attention modeling. Unlike these methods that shuffle elements across groups, we propose to use random shuffle to improve the sequential vision training for the 2-D spatial nature of image data.

## 3 METHOD

In this section, we introduce our Stochastic Layer-Wise Shuffle Regularization (SLWS) for Vision Mamba training. We briefly present the preliminaries in the following subsections for a better understanding of our algorithm, then introduce the regularization from intuition to formulation in detail.

### 3.1 PRELIMINARIES

State Space Model (SSM) (Gu et al., 2021a;b) is originally designed for modeling continuous time systems by projecting 1-D input stimulation $x(t)$ to the output signal $y(t)$ via hidden state $h(t) \in \mathbb{R}^n$. Formally, SSM is expressed with the subsequent ordinary differential equation (ODE) as follows:

$$\begin{aligned} h'(t) &= \mathbf{A}h(t) + \mathbf{B}x(t), \\ y(t) &= \mathbf{C}h(t) + \mathbf{D}x(t), \end{aligned} \quad (1)$$

where $\mathbf{A} \in \mathbb{R}^{n \times n}$ denotes the system's evolutionary matrix, with $\mathbf{B} \in \mathbb{R}^{n \times 1}$, $\mathbf{C} \in \mathbb{R}^{1 \times n}$ and $D$ are projection parameters. In a discrete system scenario, the above SSM is discreted by a timescale parameter $\mathbf{\Delta}$, transforming the expressions of $\mathbf{A}$ and $\mathbf{B}$ into their discrete equivalents $\bar{\mathbf{A}}$ and $\bar{\mathbf{B}}$. In Mamba models, such conversion is implemented with the Zero-Order Hold (ZOH) rule, which is expressed as follows:

$$\begin{aligned} \bar{\mathbf{A}} &= \exp(\mathbf{\Delta}A), \\ \bar{\mathbf{B}} &= \mathbf{\Delta}A^{-1}(\exp(\mathbf{\Delta}A - \mathbf{I})) \cdot \mathbf{\Delta}B. \end{aligned} \quad (2)$$

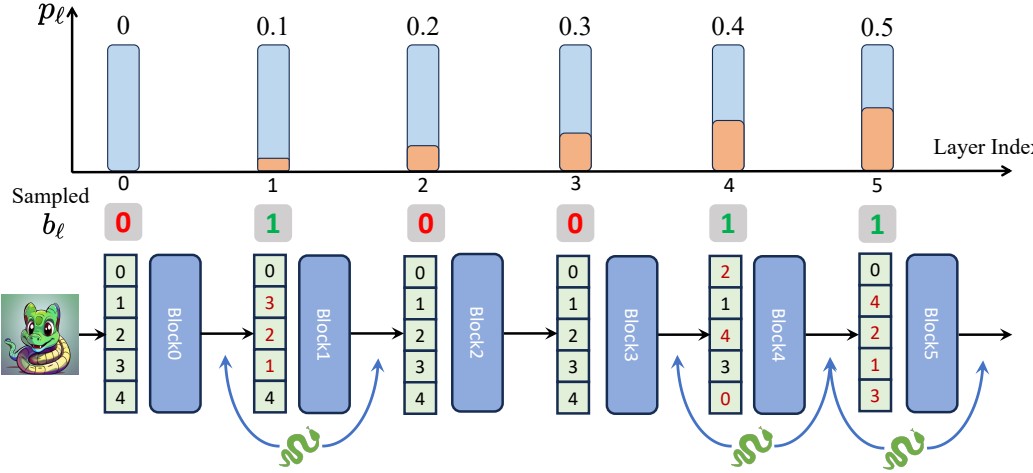

Figure 1: **Stochastic layer-wise shuffle regularization.** Higher layers are assigned with larger probabilities for shuffle regularization to enhance positional transformation invariance. $b_\ell$ is sampled according to the probability to determine to whether execute regularization. Stochastic layer-wise shuffle only includes sequence permutation and is not involved in inference.

Then, a sequential input $\{x_i\}_{i=1}^{L}$ is mapped via this discreted system to its output $\{y_i\}$ as:

$$h'_i = \bar{\mathbf{A}}h_{i-1} + \bar{\mathbf{B}}x_i,$$
$$y_i = \mathbf{C}h'_i + \mathbf{D}x_i. \tag{3}$$

Mamba (Gu & Dao, 2023) designs the $\mathbf{B}$, $\mathbf{C}$ and $\boldsymbol{\Delta}$ to be input-dependent to improve the intrinsic capacity for contextual sensitivity and adaptive weight modulation. Besides, a Selective Scan Mechanism is ensembled in for efficient computation. To this end, for a Vim (Zhu et al., 2024) block (or layer) $s_\ell$, it includes an SSM branch, whose output is multiplied by the result of another gated branch to produce the final output sequence $\boldsymbol{X}_\ell \in \mathbb{R}^{T \times D}$. Thus, the corresponding forward process is expressed in the following form:

$$\boldsymbol{X}_\ell = s_\ell\left(\boldsymbol{X}_{\ell-1}\right). \tag{4}$$

## 3.2 STOCHASTIC LAYER-WISE SHUFFLE

As formulated above, the SSM-based Mamba is initially proposed for sequence modeling but cannot be naturally adapted to 2-D image data, whose patch sequences are not casual structures. Some previous work has incorporated various scanning manners into Mamba layers to improve the spatial context perception (Zhu et al., 2024; Liu et al., 2024b; Yang et al., 2024; Li et al., 2024). Nevertheless in training, they are still stuck in the simple 1-D corner-to-corner scanning and plagued by issues such as overfitting. To improve the Vim training, we propose the stochastic layer-wise shuffle regularization according to the following intuitions:

(1) These corner-to-corner sequential scannings in SSM modules of vision models do not naturally align with the prior of capturing local neighborhood relationships and long-range global correlations.

(2) The deeper layers of a vision encoder are expected to output higher semantic-level representations, while those shallower ones provide more low-level information.

(3) Better semantic-level perception of deeper layers needs transformation invariance for patch positions, and shallower units should maintain the positional sensitivity.

(4) Adding disturbance to the basic sequential structure computing can intensify challenges associated with the visual task and thus may be beneficial for the overfitting problem.

We present the stochastic layer-wise shuffle training regularization, which introduces randomness to the corner-to-corner sequential scanning and helps to enhance the transformation invariance for

patch positions of output representations. It is a simple layer-dependent form for Vim models and is formulated as follows:

**Random Shuffle Forward Regularization.** Inspired by stochastic depth (Huang et al., 2016), we use a Bernoulli random variable $b_\ell \in \{0, 1\}$ to indicate whether the $\ell^{th}$ layer training is to be implemented with regularization. To strengthen the positional transformation invariance and intensify challenges for visual prediction task, the input token sequence $\boldsymbol{X}_{\ell-1}$ of the $\ell^{th}$ layer will be shuffled to a random order to be $\boldsymbol{X}'_{\ell-1}$ if $b_\ell = 1$, else $\boldsymbol{X}_{\ell-1}$ maintain itself. Such an operation is defined as $\pi \left( \cdot \mid b_\ell \right)$, and $\pi^{-1} \left( \cdot \mid b_\ell \right)$ or $\pi_\ell^{-1} \left( \cdot \right)$ denotes the inverse process to restore the corresponding output $\boldsymbol{X}_\ell$ to the original sequential order. Particularly, $\pi \left( \cdot \mid b_\ell \right)$ shuffles tokens obeying the simple uniform distribution. Then the forward process in Eq. (4) is reformulated as follows:

$$\boldsymbol{X}_\ell = \pi_\ell^{-1} \left( s_\ell \left( \pi \left( \boldsymbol{X}_{\ell-1} \mid b_\ell \right) \right) \right). \tag{5}$$

**Layer-Wise Probabilities Assignment.** For another, layers of Vim are assigned with different execution probabilities of training regularization. This also echoes the semantic level prior for model layers, i.e., deeper features are expected to be higher semantic. Consequently, the $\ell^{th}$ probability is designed to be an increasing function of $\ell$. In this paper, we simply take a linear form and $\ell$ starts from 0. Specifically, the probability $p_\ell$ of implementing the shuffle forward regularization for the $\ell^{th}$ layer is expressed as:

$$P \left( b_\ell = 1 \right) = \frac{\ell}{L} P_L, \tag{6}$$

where $P_L$ is a hyper-parameter of the stochastic layer-wise shuffle and will be explored in the experiment part. As we design the shuffle process to obey a discrete uniform distribution, there exists the token position transformation distribution, i.e., the probability that the $i$-th token in the $j$-th position after shuffled:

$$\begin{aligned} P \left( \boldsymbol{x}_i^\ell \Rightarrow \boldsymbol{x}_j^{'\ell} \right) &= \frac{1}{L+1} P \left( b_\ell = 1 \right) \\ &= \frac{\ell}{(L+1)L} P_L. \end{aligned} \tag{7}$$

**Efficiency Analysis.** Fig. 1 and Algorithm 1 with PyTorch functions further illustrate the SLWS algorithm for Vim training. It can be found that such a method introduces very limited extra computing costs. Particularly, the random indices generation and restoration involve the sequence length linear complexity $O(L)$ and sorting computing complexity $O(L \log L)$, respectively. As we shuffle all of the sequences in a batch with the same randomly sampled index order, the batch size does not affect the calculation of this step. Another extra operation in this regularization is gathering tensors according to the indexes of the sequence dimension, which involves

---

**Algorithm 1** Layer-Wise Shuffle forward

**Require:** token sequence $\boldsymbol{X}_{\ell-1} \in \mathbb{R}^{B \times T \times D}$,
     layer $s_\ell$, probability $p_\ell$, training flag $F$
**Ensure:** token sequence $\boldsymbol{X}_\ell$
  1: # this layer is trained with regularization
  2: **if** $F$ and rand(1) $< p_\ell$ **then**
  3:      shuffle_indices = randperm(T).expand(B, 1, D)
  4:      restore_indices = argsort(shuffle_indices, dim=1)
  5:      $\boldsymbol{X}'_{\ell-1}$ = gather($\boldsymbol{X}_{\ell-1}$, 1, shuffle_indices)
  6:      $\boldsymbol{X}'_\ell = s_\ell(\boldsymbol{X}'_{\ell-1})$
  7:      $\boldsymbol{X}_\ell$ = gather($\boldsymbol{X}'_\ell$, 1, restore_indices)
  8: **else**
  9:      # inference or trained without regularization
10:      $\boldsymbol{X}_\ell = s_\ell(\boldsymbol{X}_{\ell-1})$
11: **end if**
12: Return: $\boldsymbol{X}_\ell$

---

$O(L)$ complexity for a sequence. Therefore, the proposed Stochastic Layer-Wise Shuffle regularization only introduces $O(L \log L)$ computing complexity totally. Ablation results in Sec. 4.3 echo the limited training efficiency decrease as well.

Overall, our proposed stochastic layer-wise shuffle algorithm fulfills some advantages:

(1) The layer-dependent probability assignment and token shuffle operations are intuitive for Vision Mamba to enhance the modeling of non-casual 2-D visual data.

(2) As a training regularization, it is plug-and-play without changing the model architecture, which will be dumped in inference, and thus will not affect the application efficiency.

(3) It raises the task complexity for visual prediction to overcome overfitting but does not bring heavy extra computation as it only introduces a few complexities, thus is efficient.

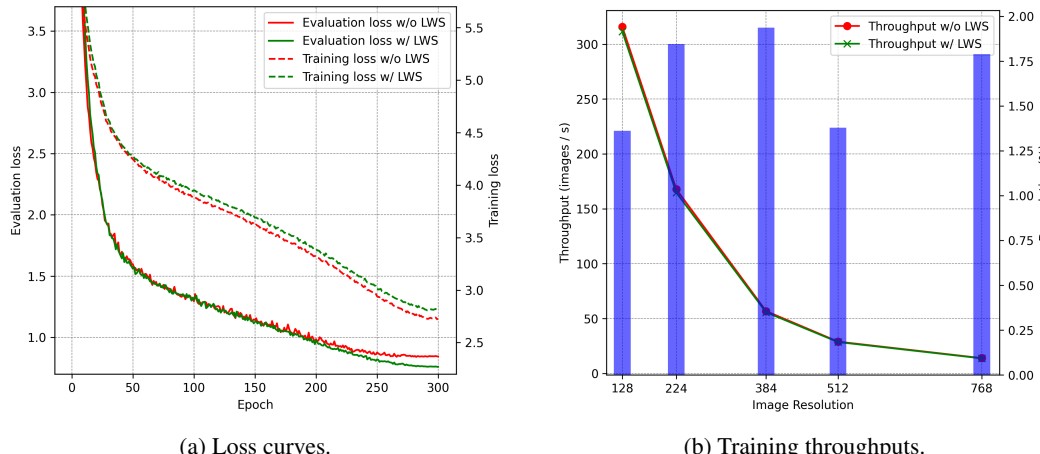

(a) Loss curves.  (b) Training throughputs.

Figure 2: **(a) Training and evaluation loss for 300 epochs middle-size Vims.** When equipped with SLWS, the model finally showcases lower evaluation loss and larger training loss. This implies that SLWS is effective for improving the overfitting problem. **(b) Training throughput change for middle-size Vims under different input resolutions.** SLWS only has very limited degradation ($< 2\%$) on training throughput.

## 4 EXPERIMENTS

In this section, we conduct comprehensive experiments to evaluate the stochastic layer-wise shuffle regularization for improving Vim training. We explored and compared the performance of different models in classification and dense prediction tasks, but also studied the algorithm properties in depth with ablations in the following subsections.

| Model | #Depth | #Dim | #Param. | #GFlops. |
|---|---|---|---|---|
| Small | 24 | 384 | 7M | 4.3 |
| Middle | 32 | 576 | 74M | 12.7 |
| Base | 24 | 768 | 98M | 16.9 |
| Large1 | 40 | 1024 | 284M | 49.8 |
| Large2 | 48 | 1024 | 340M | 59.7 |

Table 1: Configurations of models (when only one `[CLS]` token accounted) in different size.

### 4.1 IMPLEMENTATION SETTINGS

Following the common step, we train Vision Mamba models from scratch on the ImageNet-1K (Deng et al., 2009) that contains 1.28M training samples in a supervised style and evaluate them with the DeiT protocols (Touvron et al., 2021). Specifically, we take four different size models in this section, which are described in Table 1. The middle and base-size models are trained for 300 epochs with a 2048 batch size, while the Large1 is trained for 200 epochs with a 1024 batch size. We use AdamW optimizer (Loshchilov & Hutter, 2019) with selecting {20,30} epochs warmup, a cosine learning rate schedule and a 5e-4 initial basic learning rate scaled by 512. The betas and weight decay rate of AdamW are set as (0.9, 0.95) and 0.1, respectively. Mixup (Zhang et al., 2018a), Cutmix (Yun et al., 2019), Random erasing and Rand augment (Cubuk et al., 2020) are used for data augmentations. We also utilize BFloat16 precision following exiting settings for training stability. Exponential Mean Average (EMA) with a decay rate of 0.9999 classification results are reported. Besides, the drop path rate and shuffle rate $P_L$ for middle and base-size models are {0.5,0.5} while are {0.7,0.6} for ShuffleMamba-L1, respectively. Following the VideoMamba (Li et al., 2024) classification setting, we place a `[CLS]` token at the beginning of token sequences to provide classification features. For the "reg" version training, we follow Mamba-Reg (Wang et al., 2024) to perform a prefix 128 resolution pre-training (Touvron et al., 2019; 2022) and then fine-tuning along with adding same numbers of register tokens to the model.

### 4.2 RESULTS AND ANALYSIS

**Classification** Classification results on ImageNet-1K are reported in Table 2. We mainly focus on those sizes that are inferior in previous studies, i.e., middle, base, and large-size models. It can be seen that SSM-based models show competitive or better performance under similar model sizes. When compared to the ViT family (Dosovitskiy et al., 2021; Touvron et al., 2021), our

Table 2: ImageNet-1K classification comparison. All results are obtained under $224 \times 224$ resolution training except for register models. Our ShuffleMamba results are highlighted in blue .

| Arch. | Method | EMA | Distill. | Param. | FLOPs | Acc. (%) |
|---|---|---|---|---|---|---|
| | *Hierarchical* | | | | | |
| CNN | RegNetY-4G(Radosavovic et al., 2020) | | | 21M | 4G | 80.0 |
| | RegNetY-8G (Radosavovic et al., 2020) | | | 39M | 8G | 81.7 |
| | RegNetY-16G(Radosavovic et al., 2020) | | | 84M | 16G | 82.9 |
| | ConvNeXt-T(Liu et al., 2022b) | | | 29M | 4.5G | 82.1 |
| | ConvNeXt-S(Liu et al., 2022b) | | | 50M | 8.7G | 83.1 |
| | ConvNeXt-B(Liu et al., 2022b) | | | 89M | 15.4G | 83.8 |
| Trans. | Swin-T(Liu et al., 2021) | | | 28M | 4.6G | 81.3 |
| | Swin-S(Liu et al., 2021) | | | 50M | 8.7G | 83.0 |
| | Swin-B(Liu et al., 2021) | | | 88M | 15.4G | 83.5 |
| SSM | VMamba-T(Liu et al., 2024b) | ✓ | | 31M | 4.9G | 82.5 |
| | VMamba-S(Liu et al., 2024b) | ✓ | | 50M | 8.7G | 83.6 |
| | VMamba-B(Liu et al., 2024b) | ✓ | | 89M | 15.4G | 83.9 |
| | *Non-Hierarchical* | | | | | |
| CNN | ConvNeXt-S(Liu et al., 2022b) | | | 22M | 4.3G | 79.7 |
| | ConvNeXt-B(Liu et al., 2022b) | | | 87M | 16.9G | 82.0 |
| Trans. | DeiT-S | | | 22M | 4.6G | 79.8 |
| | DeiT-B(Touvron et al., 2021) | | | 87M | 17.6G | 81.8 |
| | DeiT-B(Touvron et al., 2021) | | ✓ | 87M | 17.6G | 81.9 |
| | ViT-B (MAE sup.)(He et al., 2022) | | | 87M | 17.6G | 82.1 |
| | ViT-B (MAE sup.)(He et al., 2022) | ✓ | | 87M | 17.6G | 82.3 |
| | ViT-L (MAE sup.)(He et al., 2022) | | | 309M | 191G | 81.5 |
| | ViT-L (MAE sup.)(He et al., 2022) | ✓ | | 309M | 191G | 82.6 |
| SSM | Vim-S(Zhu et al., 2024) | | | 26M | 4.3G | 80.5 |
| | VideoMamba-S(Li et al., 2024) | | | 26M | 4.3G | 81.2 |
| | VideoMamba-M(Li et al., 2024) | | | 74M | 12.7G | 80.9 |
| | VideoMamba-M(Li et al., 2024) | | ✓ | 74M | 12.7G | 82.8 |
| | VideoMamba-B(Li et al., 2024) | | | 98M | 16.9G | 79.8 |
| | VideoMamba-B(Li et al., 2024) | | ✓ | 98M | 16.9G | 82.7 |
| | LocalViM-S(Huang et al., 2024) | ✓ | | 28M | 4.8G | 81.2 |
| | PlainMamba-L2(Yang et al., 2024) | ✓ | | 25M | 8.1G | 81.6 |
| | PlainMamba-L3(Yang et al., 2024) | ✓ | | 50M | 14.4G | 82.3 |
| | Mamba-Reg-S(Wang et al., 2024) | | | 28M | 4.5G | 81.4 |
| | Mamba-Reg-B(Wang et al., 2024) | | | 99M | 17.8G | 83.0 |
| | Mamba-Reg-L(Wang et al., 2024) | | | 341M | 64.2G | **83.6** |
| | ShuffleMamba-S | | | 26M | 4.3G | 81.2 |
| | ShuffleMamba-M | | | 74M | 12.7G | 82.7 |
| | ShuffleMamba-M | ✓ | | 74M | 12.7G | 82.8 |
| | ShuffleMamba-B | | | 98M | 16.9G | 82.6 |
| | ShuffleMamba-B | ✓ | | 98M | 16.9G | 82.7 |
| | ShuffleMamba-Reg-B | | | 99M | 17.8G | **83.1** |
| | ShuffleMamba-L1 | | | 284M | 49.8G | 82.9 |
| | ShuffleMamba-L1 | ✓ | | 284M | 49.8G | 82.9 |
| | ShuffleMamba-Reg-L2 | | | 341M | 64.2G | **83.6** |
| | *256×256 Test* | | | | | |
| | Mamba-Reg-B(Wang et al., 2024) | | | 99M | 22.9G | 83.0 |
| | Mamba-Reg-L(Wang et al., 2024) | | | 341M | 82.4G | 83.2 |
| | ShuffleMamba-M | | | 74M | 16.5G | 82.8 |
| | ShuffleMamba-M | ✓ | | 74M | 16.5G | 83.0 |
| | ShuffleMamba-B | | | 98M | 22.0G | 82.9 |
| | ShuffleMamba-B | ✓ | | 98M | 22.0G | 83.0 |
| | ShuffleMamba-Reg-B | | | 98M | 22.9G | **83.2** |
| | ShuffleMamba-L1 | | | 284M | 49.8G | 83.1 |
| | ShuffleMamba-L1 | ✓ | | 284M | 49.8G | 83.2 |
| | ShuffleMamba-Reg-L2 | | | 341M | 82.4G | **83.6** |

ShuffleMamba-B has a 0.4% higher point than the supervised trained ViT-B in MAE work (He et al., 2022). ShuffleMamba-B also achieves a 0.8% accuracy higher than DeiT-B trained with the distillation technique. On the other hand, when equipped with the multi-stage training scheme and registers like (Wang et al., 2024), both Mamba-Reg and our ShuffleMamba get state-of-the-art performance among SSM-based models. Our ShuffleMamba-Reg has a slight advantage compared to Mamba-Reg. In addition, hierarchical Tansformers and SSM-based models show better classification performance.

When generalized to 256×256 test resolution (position embeddings are processed by bicubic interpolation), our ShuffleMamba models exhibit general improvements to higher testing resolution and reach the state-of-the-art place, indicating that 256×256 is included in the effective receptive fields (ERF) of our ShuffleMamba. Our ShuffleMamba-Reg models showcase a significant margin to Mamba-Reg up to 0.4%. This also confirms our basic motivation, like layer-wise semantic hypothesis and positional sensitivity for improving vision Mamba models beyond overfitting.

It is also worth noting that only Mamba-Reg and ShuffleMamba can scale the Vim model to the large size (around 300M parameters) in supervised training up to now. Thanks to our plug-and-play SLWS technology, we successfully scale up vanilla Vim with or without the need for registers.

Table 3: **Semantic segmentation results on ADE20K Val.** Computation FLOPs are measured under 512×2048 input resolution. "MS" means multi-scale test. Our ShuffleMamba results are highlighted in blue .

| type | backbone | crop size | Param. | FLOPs | mIoU | +MS |
|------|----------|-----------|--------|-------|------|-----|
| CNN | ResNet-50 | $512^2$ | 67M | 953G | 42.1 | 42.8 |
|  | ResNet-101 | $512^2$ | 85M | 1030G | 42.9 | 44.0 |
|  | ConvNeXt-B | $512^2$ | 122M | 1170G | 49.1 | 49.9 |
| Trans. | DeiT-B+MLN | $512^2$ | 144M | 2007G | 45.5 | 47.2 |
|  | ViT-B | $512^2$ | 127M | - | 46.1 | 47.1 |
|  | ViT-Adapter-B | $512^2$ | 134M | 632G | 48.8 | 49.7 |
|  | Swin-B | $512^2$ | 121M | 1170G | 48.1 | 49.7 |
| SSM | ViM-S | $512^2$ | 46M | - | 44.9 | - |
|  | Mamba-Reg-B | $512^2$ | 132M | - | 47.7 | - |
|  | Mamba-Reg-L | $512^2$ | 377M | - | 49.1 | - |
|  | ShuffleMamba-M | $512^2$ | 106M | 384G | 47.2 | 48.2 |
|  | ShuffleMamba-B | $512^2$ | 131M | 477G | 47.0 | 48.3 |
|  | ShuffleMamba-Reg-B | $512^2$ | 131M | 477G | 48.2 | 48.9 |
|  | ShuffleMamba-Reg-Adapter-B | $512^2$ | 145M | 1428G | **49.3** | **50.1** |
|  | ShuffleMamba-L1 | $512^2$ | 320M | 1168G | 48.8 | 49.9 |
|  | ShuffleMamba-Reg-L2 | $512^2$ | 376M | 1373G | **49.4** | **50.1** |

**Semantic Segmentation** To evaluate the capabilities of our ShuffleMamba in dense prediction task, we choose the semantic segmentation task and experiment on the commonly used ADE20K benchmark that contains 20K training samples. A UperNet (Xiao et al., 2018) head is built upon the ShuffleMamba backbone trained on ImageNet-1K. Following the common settings (Chen et al., 2023; Yang et al., 2024; Wang et al., 2024), we use an Adam optimizer with 0.01 weight decay and a polynomial learning rate schedule. All the models are trained for 160K iterations with batch size 16. The learning rates of the base and large-size models are set as 6e-5 and 3e-5, respectively. The [CLS] and register tokens are discarded in the segmentation task.

The mIoU results in single-scale and multi-scale testing are listed in Table 3. Representative CNN, Transformer and non-hierarchical SSM-based backbones are taken into account. With the SLWS regularization, the ShuffleMamba pre-trained models demonstrate superior performance. Our base-size model with registers outperforms ViT-B by a significant margin and the corresponding Mamba-Reg without SLWS training. When equipped with the multi-scale Adapter (Chen et al., 2023), the ShuffleMamba-Reg-Adapter-B model exhibits a further 1.6 points advantage compared to Mamba-

Reg-B and 0.5% higher than ViT-Adapter-B. Additionally, our ShuffleMamba-Reg-L2 gets the state-of-the-art accuracy on single and multi-scale test over the listed backbones in different types.

Table 4: **Object detection and instance segmentation results** using Mask R-CNN on MS COCO with $1\times$ schedule. All the listed SSM-based models use Adapter (Chen et al., 2023) structure to compute multi-scale features. FLOPs are calculated with input size $1280\times800$. Our ShuffleMamba results are highlighted in blue . Gray fonts indicate the models pre-trained on ImageNet-21K.

| type | backbone | Param. | FLOPs | $AP^b$ | $AP^b_{50}$ | $AP^b_{75}$ | $Ap^m$ | $AP^m_{50}$ | $Ap^m_{75}$ |
|------|----------|--------|-------|--------|-------------|-------------|--------|-------------|-------------|
| CNN | ConvNeXt-B | 108M | 486G | 47 | 69.4 | 51.7 | 42.7 | 66.3 | 46 |
| Trans. | Swin-B | 107M | 496G | 46.9 | - | - | 42.3 | - | - |
| | ViT-B | 114M | - | 42.9 | 65.7 | 46.8 | 39.4 | 62.6 | 42.0 |
| | ViT-L | 337M | - | 45.7 | 68.9 | 49.4 | 41.5 | 65.6 | 44.6 |
| | ViT-Adapter-B | 120M | - | 47 | 68.2 | 51.4 | 41.8 | 65.1 | 44.9 |
| | ViT-Adapter-L | 348M | - | 48.7 | 70.1 | 53.2 | 43.3 | 67.0 | 46.9 |
| SSM | PlainMamba-L3 | 79M | 696G | 46.8 | 68 | 51.1 | 41.2 | 64.7 | 43.9 |
| | ShuffleMamba-M | 103M | 564G | 46.8 | 68.8 | 50.7 | 41.8 | 65.6 | 44.8 |
| | ShuffleMamba-Reg-B | 131M | 726G | **47.7** | **69.7** | **51.8** | **42.6** | **66.7** | **45.8** |
| | ShuffleMamba-Reg-L2 | 383M | 1734G | **48.9** | **70.8** | **53.4** | **43.6** | **67.4** | **47.0** |

**Object Detection and Instance Segmentation** In this subsection, we also implement downstream object detection and instance segmentation tasks following previous work to evaluate our Shuffle-Mamba. The Mask R-CNN (He et al., 2017) structure is adopted with $1\times$ schedule for 12-epoch fine-tuning. We utilize the commonly used settings in previous work (Liu et al., 2021) and compare to different-type backbones. To compute the multi-scale features to fit the FPN network structure, we use the Adapter setup following (Yang et al., 2024; Chen et al., 2023).

The detection and instance segmentation results on the COCO dataset are reported in Table 4. It can be seen that our middle-size model is on par with the corresponding CNN and Transformer model, while the base-size model with registers outperforms ViT-Adapter-B and ConvNext-B by 0.7 points $AP^b$. Besides, our ShuffleMamba-Reg-L2 can achieve the state-of-the-art $AP^b$ and $AP^m$ among all the listed models and even be better than the ViT-Adapter-L and ViT-L trained on ImageNet-21K that is 10 times larger than our adopted ImageNet-1K. These downstream results consistently demonstrate the superiority brought by the proposed SLWS regularization.

### 4.3 ABLATION STUDIES

In this subsection, we ablate or change settings in the stochastic layer-wise shuffle regularization to investigate the effects and provide in-depth studies of this algorithm. Middle-size vanilla Vision Mamba models are adopted by default for experiments. Unless otherwise stated, the corresponding settings are the same as those in Sec. 4.1.

**SLWS is effective for mitigating overfitting.** One of the key motivations of our stochastic layer-wise shuffle regularization is to overcome the overfitting issue that prevents previous work to scaling Vim up. Fig. 2a shows the evaluation and training loss comparisons. We can observe that the model trained with SLWS finally has lower evaluation loss and higher training loss, while the ablated one tends to overfit with lower training loss but a higher evaluation error rate. This confirms the correctness of SLWS to add disturbance for sequential perception training to raise the task complexity for Vim. The results in Table 5 further suggest the effectiveness of mitigating overfitting. Specifically, though refining the training recipe in Vim and VideoMamba (80.9% with base model) can help model learning, our SLWS can bring a further 0.9% gain w.r.t. ImageNet-1K accuracy.

**SLWS has a negligible impact on training throughput.** The proposed SLWS plays a role in training for input and output sequences of a mamba block, where the efficiency has been analyzed in the former Sec 3.2. We conduct experiments with different commonly adopted training image sizes to evaluate the effect on throughput for further exploration. Fig. 2b exhibits training throughout under $128\times128$ resolution to $768\times768$ and the corresponding percentage of degradation when exploiting SLWS. It can be seen that SLWS only causes lower than 2% throughput degradation among this

Table 5: **Ablations of probability settings.** Our default setup is highlighted in blue . $P_L = 0$ indicates the model degenerates to vanilla Vim (trained with improved recipe except using SLWS).

| Probability assignment | $P_L$ | Acc. (%) |
|---|---|---|
| Layer-Dependent | 0.4 | 82.3 |
| | 0.5 | 82.7 |
| | 0.6 | 82.4 |
| | 0.7 | 82.4 |
| Constant | 0 | 81.8 |
| | 0.1 | 81.5 |
| | 0.4 | 81.1 |

Table 6: **Ablation study of [CLS] token in shuffle regularization.** We shuffle the total sequence including [CLS] token by default, which is beneficial for the classification performance of different size models.

| model | shuffle w/ [CLS] token | Acc. |
|---|---|---|
| Middle | × | 82.6 |
| | ✓ | 82.7 |
| Base | × | 82.6 |
| | ✓ | 82.6 |
| Large1 | × | 82.8 |
| | ✓ | 82.9 |

range of input sizes. Therefore, SLWS is a simple but effective and efficient training regularization for Vim.

**Layer-wise probability assignment is necessary.** The layer-wise dependent probability is a key component for the SLWS design, which introduces the semantic level prior to different layers. We list results in the context of different probability assignment settings in Table 5. We can see that the layer-dependent cases generally outperform the constant ones. Additionally, as shallower blocks are more sensitive to the patch positions, when all of the layers (except the input layer) are assigned with a through 0.1 and 0.4 probability, the model even shows inferior accuracy compared to the vanilla Vim. On the other hand, 0.5 is a better choice for the middle-size model among the listed values.

**Directly including [CLS] in shuffling is slightly better.** As the [CLS] token is taken as the feature for classification training, we experiment in this part to explore the effect of whether or not it is included in shuffling. The ablation results for different size models are shown in Table 6. It can be observed that including the [CLS] in shuffle is slightly better for middle and large models. Therefore, we just shuffle the whole sequence for blocks by default for code simplicity and the case of using registers is as the same.

## 5 CONCLUSION

In this paper, we propose a stochastic layer-wise shuffle regularization (SLWS) strategy for improving vanilla Vision Mamba training. Motivated by the semantic levels of different layers and the positional transformation invariance, we design SLWS to be layer-dependent. Specifically, deeper layers are assigned with larger probabilities to be regularized. On the other hand, SLWS is a plug-and-play algorithm, which does not change the model architecture but also only introduces light-cost permutation disturbance to token sequences. Ablation results demonstrate that our SLWS can effectively mitigate the overfitting problem of Vim and the reasonableness of the layer-wise strategy. Besides, SLWS is absent in inference and only causes negligible efficiency impact on training. More importantly, this simple but effective algorithm is verified on scalability to large-size models and superiority for comparing to state-of-the-art methods.

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
