# OpenReview forum: "Stochastic Layer-Wise Shuffle: A Good Practice to Improve Vision Mamba Training"
_ICLR.cc/2025/Conference — ICLR 2025 Conference Withdrawn Submission_

### Official Review · Reviewer_YKke · 2024-11-01

**Soundness:** 3
**Presentation:** 2
**Contribution:** 1
**Rating:** 1
**Confidence:** 5

**Summary:**

This paper introduces a regularization method called Stochastic Layer-Wise Shuffle (SLWS) to enhance the performance of Vision Mamba (Vim). The Vim model, when integrated with SLWS, is referred to as ShuffleMamba. The authors highlight a limitation of current plain Vim architectures in modeling local neighborhood relationships due to their corner-to-corner scanning pattern within the state space model. They propose that applying SLWS to shuffle token positions in deeper layers can improve large-scale vanilla Vim training, helping to mitigate overfitting. Experiments are conducted on ImageNet-1K classification, MS COCO detection, and ADE20K semantic segmentation tasks.

**Strengths:**

- The paper is well-organised.
- Motivation is clear and strong.
- extensive experiments are conducted on downstream vision tasks.

**Weaknesses:**

- The SLWS approach disrupts the inherent locality of image data. Stochastic shuffling of input token order risks causing significant performance degradation, particularly in dense prediction tasks where positional information is crucial. Even for classification tasks, deeper layers rely on coarse positional cues to distinguish object shapes, as evidenced in feature maps.

- Table 5 in [2] shows that a random scanning order results in lower performance compared to other scanning patterns, which contradicts the claimed effectiveness of SLWS.

- Figure 2(a) is confusing due to the dual-axis design (one axis on each side). Additionally, training errors are consistently much greater than evaluation errors along the x-axis, which appears unreasonable.

- The paper does not compare results with recent work, specifically ARM [2], which also addresses scaling ViM. At comparable model complexities, ARM-B (83.2%) outperforms ShuffleMamba-B (82.6%) on ImageNet-1K, and ARM-B (84.5%) surpasses ShuffleMamba-Reg-L2 (83.6%). Additionally, when scaled further, ARM-H with 662M parameters reaches 85.5% accuracy, whereas ShuffleMamba’s scaling potential appears to plateau at 341M parameters.

[2] Ren, Sucheng, et al. "Autoregressive Pretraining with Mamba in Vision." arXiv preprint arXiv:2406.07537 (2024).

**Questions:**

Please refer to the Weaknesses.

---

### Official Review · Reviewer_3nFh · 2024-11-05

**Soundness:** 3
**Presentation:** 3
**Contribution:** 2
**Rating:** 3
**Confidence:** 5

**Summary:**

The paper tries to address the scalability limitations of recent Vision Mamba (Vim) models, which, despite their lower computational complexity and competitive performance relative to Vision Transformers (ViTs), are constrained to approximately 80 million parameters due to overfitting issues. To overcome this, the authors propose a novel stochastic layerwise shuffle regularization technique that enables the successful scaling of non-hierarchical Vision Mamba models up to around 300 million parameters in supervised settings. The resulting ShuffleMamba models demonstrate superior performance, achieving 0.8% and 1.0% higher classification accuracy on ImageNet1k compared to similarly sized supervised ViTs without relying on auxiliary data. Additionally, these models exhibit significant enhancements in ADE20K semantic segmentation and COCO detection tasks. Notably, the proposed regularization method is plug-and-play, does not modify the existing model architecture, and is excluded during inference. Its simplicity lies in introducing random token permutations to mitigate overfitting, with an intuitive approach where deeper layer feature tokens are shuffled more frequently due to their increased semantic robustness and reduced sensitivity to patch positions. This work presents a compelling advancement in scaling Vision Mamba models, potentially broadening their applicability in various high-resolution and long-duration vision tasks.

**Strengths:**

1. the layer-wise shuffle is simple and bring no extra cost during inference.

**Weaknesses:**

1. Poor performance. Compared with Mamba-Reg, ShuffleMamba achieves worse performance. Besides, when work together with Mamba-Reg, Mamba-Reg-B makes no improvements over Mamba-Reg.
2. This paper claims that " It is still unclear how vanilla Vision Mamba (Vim) can be efficiently scaled up to larger sizes" However, some work[1, 2] has already scale Vim to large and even huge size.
3. Lack of speed comparison. Please report the inference time.

[1] Wang, F., Wang, J., Ren, S., Wei, G., Mei, J., Shao, W., ... & Xie, C. (2024). Mamba-r: Vision mamba also needs registers. arXiv preprint arXiv:2405.14858.
[2] Ren, S., Li, X., Tu, H., Wang, F., Shu, F., Zhang, L., ... & Xie, C. (2024). Autoregressive Pretraining with Mamba in Vision. arXiv preprint arXiv:2406.07537.

**Questions:**

Can you report the inference time?

---

### Official Review · Reviewer_G4qT · 2024-11-05

**Soundness:** 2
**Presentation:** 3
**Contribution:** 2
**Rating:** 5
**Confidence:** 4

**Summary:**

This paper argues that vanilla Vision Mamba models face an overfitting problem when scaling up, and proposes to overcome it by performing a layer-wise random token shuffle for regularization during training. The proposed ShuffleMambas are based on vanilla Vision Mamba and Mamba-reg in a plug-and-play manner. The token shuffling is performed with a layer-dependent probability.

ShuffleMamba shows higher training loss and lower evaluation loss with trivial degradation in terms of training throughput. Experiments show improved classification, semantic segmentation and object detection/instance segmentation results compared with baselines.

**Strengths:**

1. This paper is well-organized and easy to follow.

2. Experiments on different architectures, model sizes and downstream tasks provide a comprehensive evaluation of the proposed Stochastic Layer-Wise Shuffle.

**Weaknesses:**

1. Minor improvement compared to baselines: from the main experimental results from Table 2, Mamba-Reg-B(99M) - 83.0% v.s. ShuffleMamba-Reg-B(98M) - 83.1%, Mamba-Reg-L(341M) - 83.6% v.s. ShuffleMamba-Reg-L2(341M) - 83.6% showing minor or no performance improvement when equipped with proposed SLWS.

2. As SuffleMamba-S/M/B/L1 are based on vanilla Vision Mamba (Vim), why the Vim-M/B/L1 performance was not reported in Table 2? Considering the training and evaluation losses of Vim-M have already been reported in Figure 2, the missing experimental comparisons reduce the confidence in the proposed SLWS, especially when SLWS is claimed to relieve the overfitting issue of Vim models in this paper. I suggest the authors also include Vim-M/B/L1 as baselines to provide a more direct comparison if possible.

3. While standard training and testing image resolution are both 224x224 in this paper, the motivation for an additional 256×256 testing resolution remains unclear. There are limited words about the generalization ability to different image sizes and why ShuffleMamba has the potential to outperform on larger testing resolutions. Can the authors provide further clarification on the 256-resolution testing for better understanding of the proposed method?

**Questions:**

1. As SuffleMamba-S/M/B/L1 are based on vanilla Vision Mamba (Vim), why the Vim-M/B/L1 performance was not reported in Table 2 to provide a more direct comparison? Please refer to Weaknesses 2.

2. What is the motivation for an additional 256x256 test-time resolution? Please refer to Weaknesses 3.

---

### Official Review · Reviewer_ay2Y · 2024-11-05

**Soundness:** 3
**Presentation:** 3
**Contribution:** 2
**Rating:** 5
**Confidence:** 5

**Summary:**

This paper presents a stochastic layer-wise shuffle regularization method that successfully scales non-hierarchical visual mambas to large sizes in a supervised setting. It is experimentally verified that the proposed large-scale ShuffleMamba model outperforms a similar-sized supervised vit by 0.8% and 1.0% in classification accuracy on ImageNet1k without auxiliary data, and outperforms a similar-sized ViT on detection and segmentation tasks, respectively. Overall, this paper mitigates the overfitting problem of the large-scale vanilla vision mamba.

**Strengths:**

1,Clear motivation. Based on a token shuffling process to enhance the positional transformation invariance and a layer-dependent probability assignment according to the layer perception assumption.

2,Good performance and efficiency. Being a plug-and-play algorithm, the approach neither incurs heavy training costs nor changes the architecture of the visual mamba. Moreover, it outperforms existing visual mamba models on classification, detection, and segmentation tasks, and outperforms similarly sized ViTs.

3, well-written and clear presentation.

**Weaknesses:**

1, Inconsistent results. The number of parameters of ShuffleMamba-S in Table 1 is 7M, while it is 26M in Table 2.

2, Selective comparison. The convolution and transformer based methods that the authors compare are all before 2022, so the author's claim of superiority over a ViT model of the same size seems unreasonable to me. It is recommended to add recent methods such as [1-2] for comparison . However, from my observation, it seems that shuffle-mamba does not achieve better performance compared to convolution-based model [1] and transformer-based model [2] at the same model size scale.
Moreover, shuffle-mamba of similar size achieves significantly lower results compared to the recent VMamba (81.2 vs. 82.5).

3, Why the Stochastic Layer-Wise Shuffle training regularization algorithm can only be applied to nonhierarchical vision mamba?Purely from the algorithmic view, this is actually a disruption of token positions to reduce model overfitting. Therefore, this algorithm can also be used for hierarchical vision mamba such as VMamba and MSVMamba.

4, Are different shuffle methods explorable? For example, shuffling on a localized scale versus shuffling the whole image?

5, I did not find details of the shuffle-mamba design. The author's description of the structure of the whole model is rather vague. I think it is necessary for the authors to add that part of the description, especially since I observed that the authors actually have blank space to support the operation.


[1] InceptionNeXt: When Inception Meets ConvNeXt (CVPR2024)
[2] TransNeXt: Robust Foveal Visual Perception for Vision Transformers (CVPR2024)

**Questions:**

Please refer to the weaknesses.

---

### Note · Authors · 2024-11-13

I have read and agree with the venue's withdrawal policy on behalf of myself and my co-authors.